# Microbial Community Succession Associated with Poplar Wood Discoloration

**DOI:** 10.3390/plants11182420

**Published:** 2022-09-16

**Authors:** Xiaohua Zhang, Hao Liu, Heming Han, Bo Zhang, Cunzhi Zhang, Jian He, Shunpeng Li, Hui Cao

**Affiliations:** 1Key Laboratory of Agricultural Environmental Microbiology, Ministry of Agriculture and Rural Affairs, College of Life Sciences, Nanjing Agricultural University, Nanjing 210095, China; 2Bioengineering Technology Center, College of Tea and Food Science and Technology, Jiangsu Vocational College of Agriculture and Forestry, Nanjing 212499, China

**Keywords:** poplar wood, discoloration, successional stages, microbial composition, assembly processes

## Abstract

Microbes are common inhabitants of wood, but little is known about the relationship between microbial community dynamics during wood discoloration. This study uses simulation experiments to examine the changes in the microbial communities in poplar wood at different succession stages. The composition and structure of the microbial communities changed significantly in different successional stages, with an overarching pattern of bacterial diversity decreasing and fungal diversity increasing from the early to the late successional stages. Nevertheless, succession did not affect the composition of the microbial communities at the phylum level: Proteobacteria and Acidobacteria dominated the bacterial communities, while Ascomycota and Basidiomycota dominated the fungal communities. However, at the genus level, bacterial populations of *Sphingomonas* and *Methylobacterium*, and fungal populations of *Sphaeropsis* were significantly more prevalent in later successional stages. Stochastic assembly processes were dominant in the early successional stages for bacteria and fungi. However, variable selection played a more critical role in the assembly processes as succession proceeded, with bacterial communities evolving towards more deterministic processes and fungal communities towards more stochastic processes. Altogether, our results suggest that bacteria and fungi exhibit different ecological strategies in poplar wood. Understanding those strategies, the resulting changes in community structures over time, and the relationship to the different stages of poplar discoloration, is vital to the biological control of that discoloration.

## 1. Introduction

Poplars are mainly distributed throughout the northern hemisphere. Wild *Populus* spp. and its hybrids are widely planted due to their rapid growth [1]. The wood’s thermal insulation and aesthetic characteristics, its prolonged historical use and cultural significance, and its status as a natural and renewable product contribute to its wide use in buildings and furniture [2]. The commercial value of poplar wood is highly dependent on its material color and texture [3]. In nature, a wide range of factors (i.e., light, heat, and exposure to chemical or biological agents) can discolor this wood. The discoloration induced by bacteria and fungi colonization is the most damaging type of discoloration [4]. The discolored area of wood is often more extensive than the area of the pathogens-colonized area, affecting its surface and interior [5]. Discoloration can also herald changes in wood’s mechanical properties. Previous studies suggested that discoloring microorganisms use wood components as nutrients, altering its texture, composition, microstructure, and mechanical properties [6]. Wingfield and co-workers [7] reported that blue-stain is associated with a 1~2% reduction in the weight of pinewood, a 5% reduction in the modulus of fracture, a 1~4% compression of smooth grain, a 2~10% reduction in surface hardness, and a 15~30% reduction in toughness. Such changes can significantly compromise wood applications. Also, other studies have shown that the changes in wood’s texture, microstructure, and mechanical properties interact with the development, growth, and composition of the microbial colonies [8].

The primary wood-discoloring microbes are grouped in wood-decay fungus, molds, and stain fungus [8]. Wood-decay fungus belongs to the Basidiomycetes and Ascomycetes phylum and includes species associated with brown rots, white rots, and soft rots [9]. Molds belong to the Deuteromycetes phyla and include species such as *Aspergillus niger*, *Penicillium citrinum*, and *Trichoderma atroviride* (synonyms: *Lasiodiplodia theobromae and Diplodia natalensis*) [10]. Stain fungi belong to the Ascomycetes phyla, and include species such as *Botryodiplodia theobromae*, *Fusarium moniliforme*, and *Alternaria alternata* [11]. Over the last years, wood discoloration has been intensively studied, resulting in the identification of new species of discoloration microbes (e.g., *Fusarium* sp., *Sphaeropsis sapinea*, and *Spicaria anmala*) [12,13].

Sustainable forest management programs have recently heightened the need for robust methodologies for assessing forest health, including crown condition (crown transparency and discoloration) and the prevalence and distribution of fungal and insect pests [14,15]. However, studies on the biological discoloration of poplar wood have mainly focused on the screening and control of bacteria and fungi responsible for discoloration, eschewing the role played by the dynamics of the diversity and composition of the microbial communities themselves. Overall, our understanding of the poplar wood discoloration process would be enhanced by a better knowledge of the mechanisms controlling the microbial community assembly, allowing prevent discoloration by controlling the bacteria and fungi responsible.

Therefore, in this study, we investigated the structure of the microbial communities at different successional stages using simulated experiments with poplar wood samples stored for 0, 1, 2, 3, and 4 months, respectively. The diversity and composition of bacterial and fungal communities were evaluated using high-throughput sequencing. The objectives were: (1) to investigate how the composition of the bacterial and fungal communities evolves during succession, as well as the microbes that may be associated with wood discoloration; (2) the differences in how bacterial and fungal communities change during succession; (3) the variation in microbial community assembly through the course of the succession process. We hypothesized that: (1) the composition and diversity of bacterial and fungal communities would clearly shift during succession, with greater changes occurring in bacterial communities than in fungal communities, because bacteria can act as primary and secondary wood colonizers, and provide other microorganisms’ nutrient requirements. (2) Bacterial and fungal communities have different community assembly mechanisms, because bacteria and fungi have different growth strategies during succession. Our results will allow a better understanding of the microbial community during poplar discoloration, which is beneficial in providing insight into the prevention, and control of this disease.

## 2. Results

### 2.1. Changes in Microbial Diversity with Successional Stages

The alpha-diversity of the bacterial and fungal communities in the discolored poplar significantly varied across the five successional stages (Figure 1). Shannon and Chao1 indices showed a reduction in bacterial diversity from S0 to S4. In contrast, Chao1 showed no significant differences between successional stages for the fungal community, increasing from S0 to S4.

### 2.2. Changes in Microbial Community Composition with Successional Stages

The community composition of bacteria and fungi (Adonis R = 0.732, *p* = 0.001 and Adonis R = 0.795, *p* = 0.001, respectively) significantly differed between the successional stages (Figure 2). Proteobacteria, Actinobacteriota, and Bacteroidota were the most abundant phyla at all successional stages. The relative abundance of Bacteroidota, Firmicutes, and Acidobacteriota significantly changed between successional stages (Figure 3A). Specifically, the relative abundance of Actinobacteriota and Firmicutes increased through the course of the successional stages (ranging from 21.61% to 37.37%, and 1.25% to 7.28%). In contrast, the relative abundance of Bacteroidota and Chloroflexi decreased (ranging from 12.69% to 5.23%, and 6.41% to 3.36%).

The relative abundance of fungal phyla also varied (Figure 3B). The highest value of Ascomycota occurred at S0 (90.73%), and the lowest was at S4 (27.58%). In contrast, the highest value of Basidiomycota occurred at S4 (55.57%) and the lowest at S0 (0.39%).

The Bray Curtis dissimilarity showed that bacterial communities had lower dissimilarity at the earlier successional stages than at the later ones (Appendix A). S2 was the most dissimilarity successional stage for fungal communities.

### 2.3. Statistically Significant Differences in the Abundance of Bacterial and Fungal Communities between Successional Stages

Group comparison analyses were used to identify the OTUs influenced by the successional stage. Here, OTUs that exhibited a significant increase in relative abundance were referred to as “Enriched OTUs”, and those that showed a significant decrease were referred to as “Depleted OTUs”. In the bacterial community, the most significant changes occurred in the S2–S3 group, with 1418 enriched and 919 depleted OTUs (Figure 4A). In the fungal community, the most significant changes occurred in the S3–S4 group, with 45 enriched and 43 depleted OTUs (Figure 4B).

The Cladogram (Figure 5) shows the phylogenetic distribution of bacterial and fungal lineages significantly associated with samples from different successional stages. In the early stages of the succession (S1 and S2), the bacteria enriched were *Brevibacillales*, *Gammaproteobacteria*, *Clostridia*, and *Azospirillales*.

In the later stages (S3 and S4) were *Cyanobacteria*, *Actinobacteriota*, and *Acidobacteriota* (Figure 5A). In the early stages of the succession, the fungi enriched were *Ceratobasidiaceae* and *Chytridiomycota*; in the later stages were *Ustilaginomycetes*, *Botryosphaeriales*, *Hypocreales*, and *Mycosphaerellaceae* (Figure 5B).

### 2.4. The Relative Importance of Deterministic and Stochastic Processes in Shaping the Microbial Assembly with Successional Stages

The null model was used to infer the ecological processes of the microbial community assembly among the succession. The distribution of βNTI across all samples (Figure 6) showed the combined influence of stochastic and deterministic processes. The bacterial community assembly tends toward the deterministic processes (from 39.7% to 65.1%). On the contrary, the fungal community assembly process tends to develop a stochastic process (from 57.1% to 63.5%).

The Bray-Curtis metric-based results of the Raup-Crick metric showed a strong influence of selection on the assembly processes of the bacterial community (Figure 6C,E). The variable selection consistently increased from S0 to S4 stages (33% and 43%, respectively). On the contrary, the dispersal limitation consistently decreased from S0 to S4 stages (55% and 43%, respectively). For fungal communities, the effect of variable selection on community construction was more stable and varied slightly among the different stages. However, dispersal limitation and undominated processes vary widely among different stages.

## 3. Discussion

### 3.1. Variation of Microbial Community Diversity across Successional Stages

Results demonstrated that the diversity of bacterial and fungal communities was altered among succession. Bacterial community diversity increased from early to late stages, whereas fungal community diversity decreased, consistent with previous studies [16]. Furthermore, the differences in bacterial community diversity between successional stages were more significant than those in fungal community diversity. The different growth strategies of microbes may be responsible for the significantly decreased bacterial alpha-diversity index. For example, in primary succession, microbial populations live in harsh and unpredictable environments, presenting high reproduction rates and low survival, leading to a high diversity and low abundance. In contrast, in the later stages of the succession, microbial populations live in favorable and predictable environments, presenting low repopulation rates and high survival, leading to increased competitiveness and stable population size [17]. An extensive regional survey revealed that environmental factors are drivers of community diversity and composition [18]. The differences in poplar alpha-diversity in bacteria and fungi communities may be due to microorganisms with different nutritional preferences [19,20] or bacteria and fungi occupation of different niches [21].

### 3.2. Changes in Microbial Community Structure with Succession

Differential abundance analysis was used to study the differences in the microbial community and identify enriched and depleted OTUs. A more significant proportion of enriched and depleted OTUs were found at the initial stages of the bacterial community. In the fungal community, more significant alterations were found at later stages. In addition, we observed that the bacterial community underwent relatively more significant changes than the fungal community, consistent with previous studies [16,22]. Bacteria and fungi have been shown in several studies to prefer r-strategy and k-strategy groups, respectively [23,24]. Zhou and co-workers [25] observed that the r-strategy group predominated in early successional forests, and that the k-strategy group was widespread in later successional forests. This may result from faster growth and turnover rates of bacteria compared with fungi during early successional stages [26,27]. This effect also appears to exist in the succession process of poplar wood. In addition, bacteria can act as primary and secondary wood colonizers, fix nitrogen, degrade or modify wood’s chemical composition, and provide other microorganisms’ nutrient requirements [28]. Moreover, the bacterial community was more variable than the fungal community, probably due to the relatively short duration of the experiment.

We further analyzed the changing pattern of microbial taxonomic composition in five successional stages. Bacterial groups, such as Proteobacteria, Actinobacteriota, and Bacteroidota, were dominant bacterial taxa in the five successional stages, consistent with the findings of a growing number of studies in different ecosystems [29,30]. Notably, the relative abundance of two genera, *Sphingomonas* and *Methylobacterium*, belonging to Proteobacteria, increased significantly with succession (Appendix A). Previous studies have shown that *Sphingomonas* has a strong aromatic decomposition ability, degrading aromatic compounds containing methyl groups to methanol and aromatics through demethylation reaction, and can play an essential role in the decomposition of lignin [31,32]. At the same time, the released methanol act as a carbon source for *Methylobacterium*, which is beneficial to the growth of this bacteria [33]. These findings suggested that *Sphingomonas* and *Methylobacterium* play a functional role during the discoloration of poplar wood.

Ascomycota and Basidiomycota were the main phyla in the poplar for fungal communities among different successional stages, presenting significant abundance changes. This result was consistent with the findings of previous studies [34,35]. This alteration was mainly due to the significant increase of the genus *Sphaeropsis* (Ascomycota), considered a crucial contributor to wood discoloration (Appendix A) [36,37]. LEfSe analysis was also used to understand microbial communities’ variation in different succession stages. *Sphaeropsis* were enriched at the S4 stage, suggesting that this fungus plays a vital role in the discoloration of poplar wood.

### 3.3. The Different Assembly Processes Underwent by Bacterial and Fungal Communities across Successional Stages

The fundamental topic of microbial ecology is to uncover the underlying microbial assembly processes [38]. Niche theory states that deterministic factors(i.e., species traits, environmental conditions, and interactions among species) determine the structure of communities [39]. βNTI values were calculated, and RC_bray_ analysis was performed to investigate the role of neutral and niche processes.

Results showed that the stochastic processes control the bacterial assembly processes in the early stages, but the stochastic processes decrease with the succession stages, evolving from stochastic to deterministic processes. Similarly, the fungal community was controlled by stochastic processes in the early stages, but in contrast, the assembly process evolves further into stochastic processes during the later successional stages. These findings are consistent with previous studies that assumed that the initial establishment of a community is primarily determined by stochasticity [40]. For example, such studies demonstrated that microbial communities in the early stages of the succession are less similar, being dominated by taxa that can use many different resources, suggesting a strong potential influence of stochasticity [41]. More generally, if a variety of species can evolve effectively in a given environment, stochasticity is likely to dominate the early stages of community formation [42]. Furthermore, the different assemblage processes could be related to the various growth strategies of bacteria and fungi within the succession [25]. Bacterial diversity decreases with succession, while fungal diversity increases with succession. According to a recent study, highly diverse communities are dominated by stochastic processes, while low diverse communities are dominated by deterministic processes that limit community functioning, which is consistent with our findings [43].

RC_bray_ analysis results indicate that variable selection was the dominant ecological process shaping the microbial community among different successional stages, followed by dispersal limitation and undominated process. Variable selection increases with the successional stage. Previous research has shown that communities with low biomass and population size are more susceptible to drift (stochastic processes) or founder effects. However, in an established community with a saturated population or small community size, the local role of stochastic or deterministic processes may be significantly affected by dispersal influenced by the local environment [43]. Dispersal limitation has been widely accepted as an essential driver in shaping microbial community assemblages at different successional stages [44]. The Shannon and Chao1 indices of the bacterial community showed a decreasing trend from S0 to S4. In contrast, the Shannon index of the fungal community showed an increasing trend from S0 to S4, while Chao1 of fungal showed no significant differences between successional stages. These results are consistent with previous research showing that low species diversity in communities could enhance environmental selection and contribute to deterministic (selection) processes followed by dispersal limitation [43]. Another study reported that colonization of specific taxa in a specific environment enhances environmental selection [45]. Due to the regional variance in the selection environment, taxa selected in one area may not be selected in another [46]. For example, new spatially structured ecological niches emerge (e.g., anoxic pockets), and communities are driven toward a 3D architecture [47]. It is expected that increasing environmental variability in this situation will lead to differences in the composition of local communities. As a result, the relative importance of variable selection is expected to increase as succession progresses. In this study, we also noted that some functional taxa (e.g., *Sphingomonas*, *Methylobacterium*, and *Sphaeropsis*) increased significantly with the successional stage. Furthermore, the undominated process, which accounts for a relatively large proportion of the fungal assembly process, is also present throughout the succession; being difficult to predict the variation of fungal communities within the succession process [48].

## 4. Materials and Methods

### 4.1. Sample Collection and Site Description

Samples of Nanlin 95 poplar of similar age were taken from the city of Suqian (Jiangsu Province, China) and then normal stored at the Jurong wood factory (Zhenjiang, Jiangsu Province, China) The storage area of poplar samples has a subtropical monsoon climate, with abundant sunshine and rainfall (mean annual temperature, 17.58 °C; mean annual precipitation, 1260 mm). The environmental conditions for the storage of poplar wood are similar to those of recent years. Further samples were then taken from the stored wood at intervals of 0, 1, 2, 3, and 4 months, 5 sampling times in total (numbered S0, S1, S2, S3, and S4 respectively). The five-point sampling method collected circular sections of 2.5 cm diameter at positions with uniform discoloration. The entire sample circular section was wiped with a sterile degreasing cotton ball. Three replicates were taken for each sampling point. After sampling was completed, kept it in sterilized plastic zipper bags at the study location. Then transferred to the Key Laboratory of Agricultural Environmental Microbiology, Ministry of Agriculture and Rural Affairs, College of Life Sciences, Nanjing Agricultural University, Nanjing, China, via ice bag for microbiological analysis.

### 4.2. DNA Extraction and High-Throughput Sequencing

Genomic DNA was extracted using a Fast DNA™ Spin Kit (MP Biomedicals, USA), following the manufacturer’s instructions. A NanoDrop 2000 Spectrophotometer (Thermo Scientific, Wilmington, DE, USA) was used to determine the quality of the extracted DNA. The bacterial-specific V3-V4 region was amplified with the 338F (5′-ACTCCTACGGGAGGCAGCA-3′) and 806R (5′-GGACTACHVGGGTWTCTAAT-3′) The fungal-specific ITS region was amplified with the ITS1 (GTGAATCATCGARTC) and ITS2 (TCCTCCGCTTATTGAT) primer sets. The parameters for the PCR amplification were: 3 min of initial denaturation at 94 °C, followed by 24 cycles consisting of 5 s of denaturation at 95 °C, 90 s of annealing at 57 °C, 10 s of elongation at 72 °C, and a final extension at 72 °C for 5 min. Pooled PCR products were purified using the GeneJETTM Gel Extraction Kit (Thermo Scientific, Wilmington, NC, USA). Finally, GENEWIZ, Inc (South Plainfield, NJ, USA) sequenced the purified products using an Illumina Miseq platform (Illumina, San Diego, CA, USA).

### 4.3. Processing of Sequence Analysis

The QIIME pipeline (Version 1.9.1, J Gregory Caporaso, St. Louis, MO, USA, 28 June 2022) was used to process the microbiome sequences [49]. Reads shorter than 200 bp and with an average base quality score of <20 were considered low quality and omitted from further analysis. The UCHIME method was used to classify and delete chimeric sequences [50]. Sequences with 97% identity were grouped into operational taxonomic units (OTUs) [51]. An annotated OTU analysis was performed using the Mothur algorithm to find representatives of taxa and count the number of OTUs per sample in the taxonomic information [52]. Bacterial annotation followed the Silva database (https://www.arb-silva.de/ 28 June 2022), while fungal annotations followed the UNITE database (https://unite.ut.ee/ 30 June 2022).

### 4.4. Statistical and Bioinformatics Analysis

Bacterial and fungal community alpha-diversity indices (Shannon and Chao1) were calculated using the R package “vegan” [53]. Beta-diversity indices (Bray-Crust dissimilarity) were calculated using QIIME. The R “vegan” package was also used to reconstruct the bacterial and fungal community composition, allowing Principal Coordinates Analysis (PCoA) based on dissimilarity matrices.

The differential abundance of OTUs was illustrated using Volcano plots, with *p*-values adjusted according to the false discovery rate (FDR) [54]. The log2 Fold Change (log2FC) and adjusted *p*-values were calculated using the “DESeq2” R package. The “ggplot2” package was used to construct differential abundance plots for the OTUs. Statistical biomarkers between treatments were identified through Linear discriminant analysis effect size (LEfSe) measurements [55] on the Hutlab Galaxy website application (http://huttenhowe.sph.harvard.edu/galaxy/ 23 July 2022).

This study preferentially performed Null-modeling-based approaches to infer community assembly mechanisms [56]. With this approach, the community assembly mechanism can be detected by estimating the observed ecological patterns’ standard deviation compared to the randomly shuffled ecological patterns generated by the null model. Generally, variations were investigated by comparing beta-diversity metrics (βNTI values and RC_bray_). Here, the null model analysis was performed using a framework described by Stegen and co-workers [57] to classify community pairs into underlying drivers of deterministic processes (e.g., homogeneous selection and variable selection) or stochastic processes (e.g., dispersal limitation, homogeneous dispersal, and “non-dominant”). Briefly, The relative influence of dispersal limitation, homogeneous dispersal, and undominated process were quantified as a pairwise comparison between |βNTI| < 2, RC_bray_ > 0.95, |βNTI| < 2, RC_bray_ < −0.95 and |βNTI| < 2, −0.95 < RC_bray_ < 0.95, respectively [58].

## 5. Conclusions

This study investigated the variation of poplar wood microbial communities in response to different successional stages. Our results showed that the diversity and composition of microbial communities varied greatly with succession. Bacterial diversity increased, and fungal diversity decreased as the succession proceeded, indicating that bacterial and fungal communities had differential responses to succession. The relative abundance of the dominant phyla was significantly different across the five successional stages: Proteobacteria and Acidobacteria were the most common phyla of bacteria, while Ascomycota and Basidiomycota were the most common phyla of fungus. Notably, the relative abundance of *Sphingomonas*, *Methylobacterium*, and *Sphaeropsis*, significantly increased with the successional stage. The structure of microbial communities showed a separation across the successional stages. Based on the volcano plot analysis, we found that the bacteria experienced more alteration in the early stages of the succession, while the fungi occurred in the late stages. The main ecological processes for microbial community assembly are variable selection, dispersal limitation, and undominated process. Stochastic processes dominate both bacterial and fungal assembly processes in the early stages, In the course of succession, bacterial community assembly processes tend to develop towards deterministic processes, whereas, on the contrary, fungi tend to become more stochastic. In summary, it provides insights into the changes in microbial communities during poplar wood discoloration.

## Figures and Tables

**Figure 1 plants-11-02420-f001:**
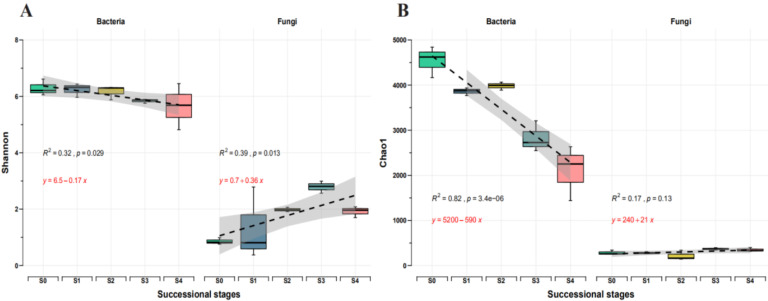
The dynamic changes of microbial alpha-diversity at different successional stages. (**A**) Shannon; (**B**) Chao1; Linear trend is the ordinary least-squares linear regressions; shaded area indicates confidence interval (0.95); S0 to S4 represent successional stages from 0 to 4, respectively.

**Figure 2 plants-11-02420-f002:**
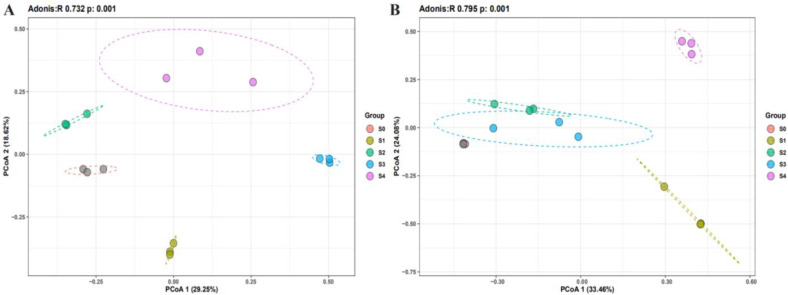
Principal coordinate analysis (PCoA) plot of the first two principal components based on bacterial (**A**) and fungal (**B**)community compositions at each successional stage.

**Figure 3 plants-11-02420-f003:**
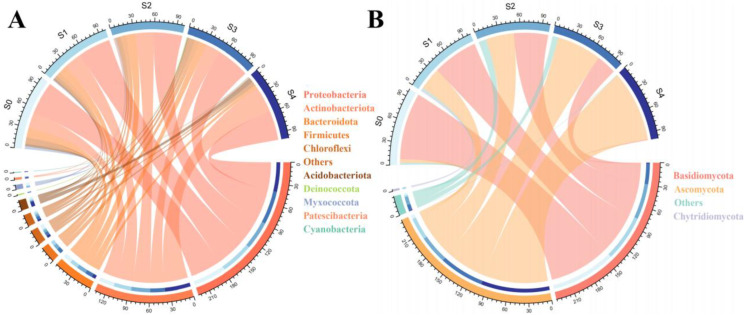
Composition of bacterial (**A**) and fungal (**B**) communities, at the phylum level, from discolored poplar wood across five successional stages. The different colors represent the various taxa.

**Figure 4 plants-11-02420-f004:**
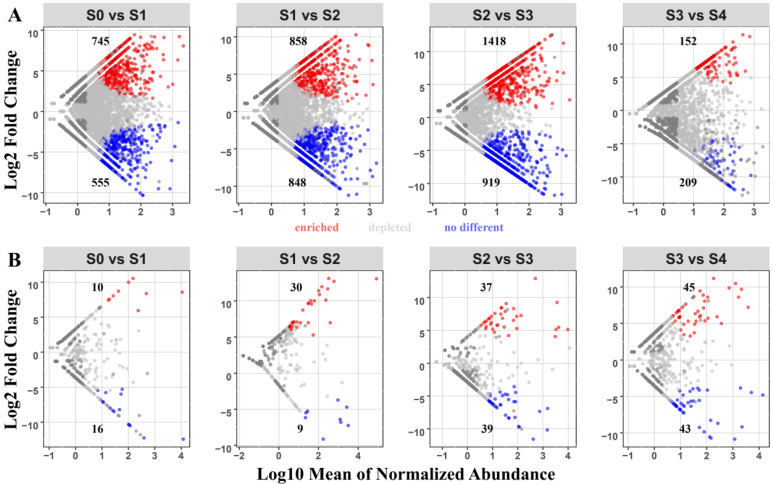
Volcano plots illustrating bacterial (**A**) and fungal (**B**) OTUs in poplar wood with increases (red) and reductions (blue) in abundance between successional stages; Each point indicates an individual OTU.

**Figure 5 plants-11-02420-f005:**
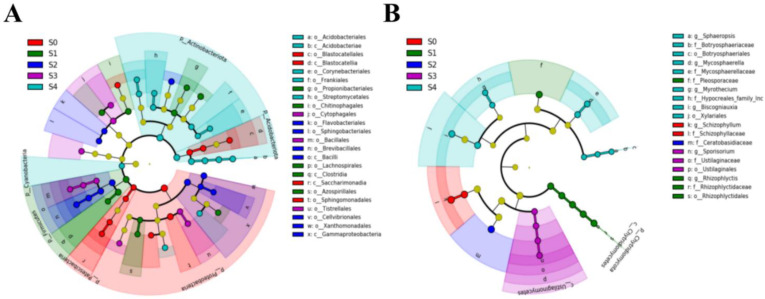
LDA Effect Size analysis of bacteria (**A**) and fungi (**B**) communities in poplar wood at different successional stages. Only effect sizes with LDA > 4 for bacterial communities, and LDA > 2 for fungal communities are shown.

**Figure 6 plants-11-02420-f006:**
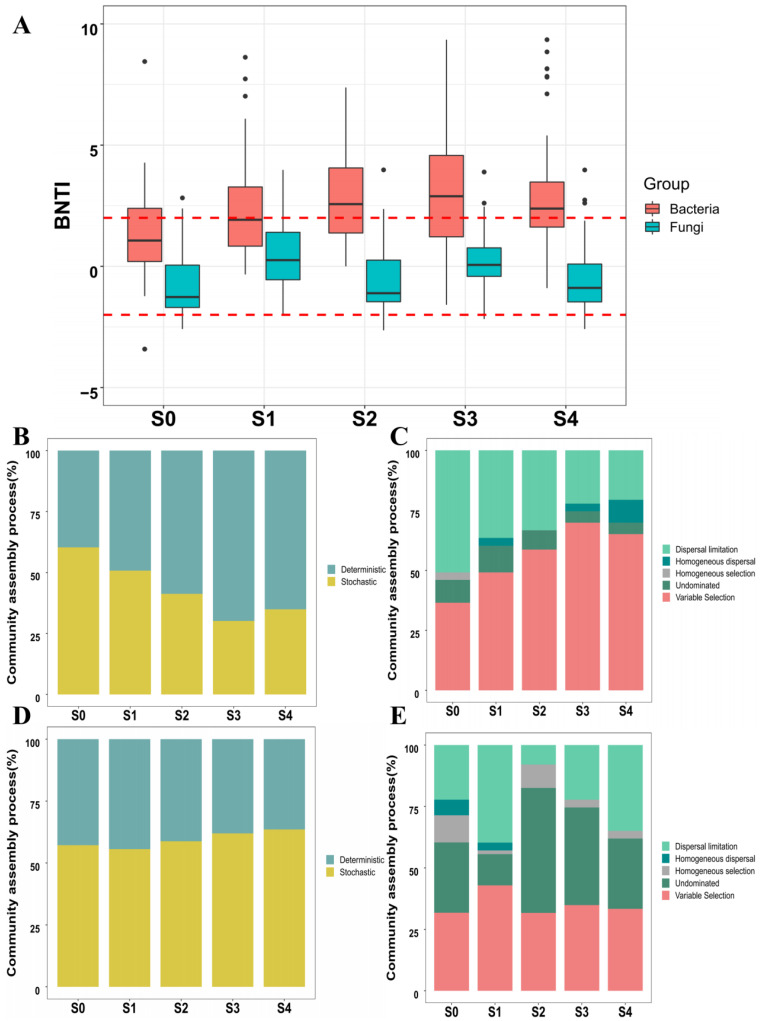
Microbial community assembly process associated with poplar wood discoloration. (**A**) βNTI results of the bacterial and fungal community; (**B**,**D**) Proportion of deterministic and stochastic processes of the bacterial and fungal community; and (**C**,**E**) RC_bray_ results of the bacterial and fungal community.

## Data Availability

Not applicable.

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
