# Peer review of "Microbial Community Succession Associated with Poplar Wood Discoloration"

_plants, 2022, doi:10.3390/plants11182420_

Round 1

Reviewer 1 Report

The study by Huang et al. is an interesting work showing that bacteria and fungi exhibit different ecological strategies in poplar wood. This study also provides basis for the development of more appropriate forest management approaches.

However, I found that the methods section is not quite clear. It is not easy to understand how the samples were obtained, and stored, and under which environmental conditions? For me it was really difficult to understand this section. Was the wood collection performed only once? When? Were the samples obtained from trees with a similar age? How can we be sure that the observed results are not exclusive to the particular conditions of the studied year? Please clarify this section.

Author Response

Point 1: The study by Huang et al. is an interesting work showing that bacteria and fungi exhibit different ecological strategies in poplar wood. This study also provides basis for the development of more appropriate forest management approaches. However, I found that the methods section is not quite clear. It is not easy to understand how the samples were obtained, and stored, and under which environmental conditions? For me it was really difficult to understand this section. Was the wood collection performed only once? When? Were the samples obtained from trees with a similar age? How can we be sure that the observed results are not exclusive to the particular conditions of the studied year? Please clarify this section. 

Response 1: Thank you for your constructive and helpful suggestion. We are sorry for the poor presentation in this section. Our samples were collected from poplar trees of similar age and then stored in a wood factory. The environmental conditions were the local conditions at that time. Subsequently, we conducted a 4-month trial in the wood factory, sampling the wood at one-month intervals, plus a control for a total of 5 times. The growth and reproduction of microbes are strongly influenced by environmental factors. we investigated the environmental conditions for this year and they did not differ much compared to recent years. Therefore, we can determine the observed results and can exclude the specific conditions of the study year. We have revised the manuscript carefully according to your suggestions. We hope these revisions will be satisfactory for publication.

Reviewer 2 Report

I didn't see the methodologies

The article "Microbial community succession associated with poplar wood discoloration" addresses an original and interesting subject and adds value to publications on the subject. The document is well written, the text is clear and easy to read and the conclusions are consistent with the data and results presented they address the main issue and meet the objectives. 

Author Response

Point 1: I didn't see the methodologies. The article "Microbial community succession associated with poplar wood discoloration" addresses an original and interesting subject and adds value to publications on the subject. The document is well written, the text is clear and easy to read and the conclusions are consistent with the data and results presented they address the main issue and meet the objectives.

 Response 1: Thank you very much for your approval of our research. Due to formatting requirements, methodologies are presented at the end of the manuscript, please check them again.

Reviewer 3 Report

As general comment the work is well written and designed with relevant results.

In general terms the topic of the article is interesting, the methodology is explicitly presented and the results reported are interesting.

The structure of the paper is correct.

In my opinion, the abstract is too general, please reframe.

The introduction chapter should end with a paragraph indicating the purposefulness of the conducted research. Authors should clearly define the purpose of the work and formulate research hypotheses.

Materials and method section is well described and correspond to the aim set out in the manuscript. The figures clearly presenting the obtained results with their appropriate interpretation. The statistical calculation methods used in the research make the obtained results reliable and provide a basis for drawing correct conclusions.

The references are sufficient and necessary.

The paper needs some editorial corrections.

I recommend the publication of this manuscript in the Plants journal after minor revisions.

Author Response

As general comment the work is well written and designed with relevant results. In general terms the topic of the article is interesting, the methodology is explicitly presented and the results reported are interesting. The structure of the paper is correct. Materials and method section is well described and correspond to the aim set out in the manuscript. The figures clearly presenting the obtained results with their appropriate interpretation. The statistical calculation methods used in the research make the obtained results reliable and provide a basis for drawing correct conclusions. The references are sufficient and necessary. The paper needs some editorial corrections. I recommend the publication of this manuscript in the Plants journal after minor revisions.

Point 1: In my opinion, the abstract is too general, please reframe. 

Response 1: Thank you for your constructive and helpful suggestion. We are sorry for the poor presentation in this section.We have rewritten this sentence.

Point 2: The introduction chapter should end with a paragraph indicating the purposefulness of the conducted research. Authors should clearly define the purpose of the work and formulate research hypotheses.

Response 2: Thank you very much for your great suggestions.In revised manuscript,We further clarified the purpose of the study and added hypotheses

Point 3: The paper needs some editorial corrections.

Response 3: Thank you very much for your valuable suggestions. We have carefully corrected the language and grammatical errors in this revision.The manuscript was also checked for clarity and accuracy of English by Academic Research Editors Ltd., a company registered in England. In the attachment, we also provide proof of editing. We believe current version of manuscript will be accepted for publication.

Reviewer 4 Report

Title: Microbial community succession associated with poplar wood discoloration

Authors: Xiaohua Zhang, Hao Liu, Heming Han, Bo Zhang, Cunzhi Zhang, Jian He, Shunpeng Li, Hui C

Reference: plants-1928065

Article type: Research

Reviewer Comments:

The manuscript plants-1928065, entitled “Microbial community succession associated with poplar wood discoloration”, studies the dynamic changes in the structure of the microbial communities in poplar wood across different succession stages.

General comments:

In the reviewer's opinion, the authors of this paper do not have English as their native tongue. A high degree of editing and revising will be necessary before the manuscript can be appropriately reviewed, as there are numerous statements whose meaning is unclear, repeated sections of the text (sometimes the same text is repeated three times), the is convoluted and hard-to-read. Also, some formatting issues must be carefully corrected (for instance, figure legends).

Therefore, due to the extension of the corrections needed, instead of presenting the suggestion by line, the reviewer presents the bulk sections of the text (suggestion in bold).

In the Result section, several sentences were removed since they were Discussion and not Results.

Specific comments:

Lines 9-15: confusing text, please consider rewrite

Line 16: please consider replacing it with

Alterations in the color of poplar wood are associated with changes in microbial communities within the wood. This study uses simulation experiments to examine the changes in the microbial communities in poplar wood at succession stages.

The composition and structure of the microbial communities significantly changed in different successional stages, with an overarching pattern of bacterial diversity decreasing and fungal diversity increasing from the early to the late successional stages. Nevertheless, succession did not affect the composition of the microbial communities at the phylum level: Proteobacteria and Acidobacteria consistently dominated the bacterial communities, while Ascomycota and Basidiomycota dominated the fungal communities. However, at the genus level, bacterial populations of Sphingomonas and Methylobacterium and fungal populations of Sphaeropsis were significantly more prevalent in their later stages. Stochastic processes were dominant in the early successional stages for bacteria and fungi. However, variable selection played a more critical role in the assembly processes as succession proceeded, with bacterial communities evolving towards more deterministic processes and fungal communities towards more stochastic processes. Altogether, our results suggest that bacteria and fungi exhibit different ecological strategies in poplar wood. Understanding those strategies, the resulting changes in community structures over time, and the relationship to the different stages of poplar discoloration, is vital to the biological control of that discoloration.

Lines 30-91: please consider replacing it with

Poplars are mainly distributed throughout the northern hemisphere. Wild Populus spp. and its hybrids are widely planted due to their rapid growth [1]. The wood's thermal insulation and aesthetic characteristics, its prolonged historical use and cultural significance, and its status as a natural and renewable product contribute to its wide use in buildings and furniture [2].

The commercial value of poplar wood is highly dependent on its material color and texture [3]. In nature, a wide range of factors (i.e., light, heat, and exposure to chemical or biological agents) can discolor this wood. The discoloration induced by bacteria and fungi colonization is the most damaging type of discoloration [4]. The discolored area of the wood is often more extensive than the area of the pathogens-colonized area, affecting its surface and interior [5]. Discoloration can also herald changes in wood's mechanical properties.

Previous studies suggested that discoloring microorganisms use wood components as nutrients, altering its texture, composition, microstructure, and mechanical properties [6]. Wingfield and co-workers [7] reported that blue-stain is associated with a 1~2% reduction in the weight in pinewood, a 5% reduction in the modulus of fracture, a 1~4% compression of smooth grain, a 2~10% reduction in surface hardness, and a 15~30% reduction in toughness. Such changes can significantly compromise wood applications [ref]. Also, other studies have shown that the changes in wood's texture, microstructure, and mechanical properties interact with the development, growth, and composition of the microbial colonies [8].

The primary wood-discoloring microbes are grouped in wood-decay fungus, molds, and stain fungus [10]. Wood-decay fungus belongs to the Basidiomycetes and Ascomycetes phylum and includes species associated with brown rots, white rots, and soft rots [ref]. Molds belong to the Deuteromycetes phyla and include species such as Aspergillus niger, Penicillium citrinum, and Trichoderma atroviride (synonyms: Lasiodiplodia theobromae and Diplodia natalensis) [ref]. Stain fungi belong to the Ascomycetes phyla, and include species such as Botryodiplodia theobromae, Fusarium moniliforme, and Alternaria alternata [ref]. Over the last years, wood discoloration has been intensively studied, resulting in the identification of new species of discoloration microbes (e.g., Fusarium sp., Sphaeropsis sapinea, and Spicaria anmala)[11, 12].

Sustainable forest management programs have recently heightened the need for robust methodologies for assessing forest health, including crown condition (crown transparency and discoloration) and the prevalence and distribution of fungal and insect pests [13, 14]. However, studies on the biological discoloration of poplar wood have mainly focused on the screening and control of bacteria and fungi responsible for discoloration, eschewing the role played by the dynamics of the diversity and composition of the microbial communities themselves. Overall, our understanding of the poplar wood discoloration process would be enhanced by a better knowledge of the mechanisms controlling the microbial community assembly, allowing prevent discoloration by controlling the bacteria and fungi responsible.

Therefore, in this study, we investigated the structure of the microbial communities at different successional stages using simulated experiments with poplar wood samples stored for 0, 1, 2, 3, and 4 months. The diversity and composition of bacterial and fungal communities were evaluated using high-throughput sequencing.

Our results will allow a better understanding of the microbial community during poplar discoloration, which is beneficial in providing insight into the prevention, and control of this disease.

Lines 92-199: please consider replacing it with

2. Results

2.1 Changes in microbial diversity with successional stages

The alpha-diversity of the bacterial and fungal communities in the discolored poplar significantly varied across the five successional stages (Fig. 1). Shannon and Chao1 indices showed a reduction in bacterial diversity from S0 to S4. In contrast, Chao1 showed no significant differences between successional stages for the fungal community, increasing from S0 to S4.

Figure 1 The dynamic changes of microbial alpha-diversity at different successional stages. (A) Shannon; (B) Chao1; Linear trend is the ordinary least-squares linear regressions; shaded area indicates confidence interval (0.95); S0 to S4 represent successional stages from 0 to 4, respectively. 

2.2 Changes in microbial community composition with successional stages

The community composition of bacteria and fungi (Adonis R = 0.732, p = 0.001 and Adonis R = 0.795, p = 0.001, respectively) significantly differed between the successional stages (Fig. 2). Proteobacteria, Actinobacteriota, and Bacteroidota were the most abundant phyla at all successional stages. The relative abundance of Bacteroidota, Firmicutes, and Acidobacteriota significantly changed between successional stages (Fig. 3A). Specifically, the relative abundance of Actinobacteriota and Firmicutes increased through the course of the successional stages (ranging from 21.61% to 37.37% and 1.25% to 7.28%). In contrast, the relative abundance of Bacteroidota and Chloroflexi decreased (ranging from 12.69% to 114 5.23% and 6.41% to 3.36%).

Figure 2. Principal coordinate analysis (PCoA) plot of the first two principal components based on bacterial (A) and fungal (B)community compositions at each successional stage.

The relative abundance of fungal phyla also varied (Fig. 3B). The highest value of Ascomycota occurred at S0 (90.73%) and the lowest at S4 (27.58%). In contrast, the highest value of Basidiomycota occurred at S4 (55.57%) and the lowest at S0 (0.39%). The Bray Curtis dissimilarity showed that bacterial communities had lower dissimilarity at the earlier successional stages than at the later ones (Fig. S1). S2 was the most dissimilarity successional stage for fungal communities.

Figure 3. Composition of the bacterial (A) and fungal communities (B), at the phylum level, from discolored poplar wood across five successional stages. The different colors represent the various taxa.

2.3 Statistically significant differences in the abundance of bacterial and fungal communities between successional stages

Group comparison analyses were used to identify the OTUs influenced by the successional stages. OTUs that exhibited a significant increase in relative abundance were referred to as "Enriched OTUs," and those that showed a significant decrease were referred to as "Depleted OTUs." In the bacterial community, the most significant changes occurred in the S2–S3 group, with 1418 enriched and 919 depleted OTUs (Fig. 4A). In the fungal community, the most significant changes occurred in the S3–S4 group, with 45 enriched and 43 145 depleted OTUs (Fig. 4B). 

Figure 4. Volcano plots illustrating bacterial (A) and fungal OTUs (B) in poplar wood with increases (red) and reductions (blue) in abundance between successional stages. Each point indicates an individual OTU.

Figure 5. LDA Effect Size analysis of bacteria (A) and fungi (B) communities in poplar wood at different successional stages. Only effect sizes with LDA > 4 for bacterial communities and LDA > 2 for fungal communities are shown.

The cladogram (Fig. 5) shows the phylogenetic distribution of bacterial and fungal lineages significantly associated with samples from different successional stages. In the early stages of the succession (S1 and S2), the bacteria enriched were Brevibacillales, Gammaproteobacteria, Clostridia, and Azospirillales; in the later stages (S3 and S4) were Cyanobacteria, Actinobacteriota, and Acidobacteriota (Fig. 5A). In the early stages of the succession, the fungi enriched were Ceratobasidiaceae and Chytridiomycota; in the later stages were Ustilaginomycetes, Botryosphaeriales, Hypocreales, and Mycosphaerellaceae (Fig. 5B).

2.4 The relative importance of deterministic and stochastic processes in shaping the microbial assembly with successional stages

The null model was used to infer the ecological processes of the microbial community assembly among the succession. 

The distribution of βNTI across all samples (Fig. 6) showed the combined influence of stochastic and deterministic processes. The bacterial community assembly tends toward the deterministic processes (from 39.7% to 65.1%). On the contrary, the fungal community assembly process tends to develop toward a stochastic process (from 57.1% to 63.5%).

The Bray-Curtis metric-based results of the Raup-Crick metric showed a strong influence of selection on the assembly processes of the bacterial community (Fig. 6C, 6E). The variable selection consistently increased from S0 to S4 stages (33% and 43%, respectively). On the contrary, the dispersal limitation consistently decreased from S0 to S4 stages (55% and 43%, respectively). For fungal communities, the effect of variable selection on community construction was more stable and varied slightly among the different stages. However, dispersal limitation and undominated processes vary widely among different stages.

Figure 6. Microbial community assembly process associated with poplar wood discoloration. (A) βNTI results of the bacterial and fungal community; (B and D) Proportion of deterministic and stochastic processes of the bacterial and fungal community; and (C and E) RCbray results of the bacterial and fungal community.

Lines 200-315: please consider replacing it with

3. Discussion

3.1 Variation of microbial community diversity across successional stages

Results demonstrated that the diversity of bacterial and fungal communities was altered among succession. Bacterial community diversity increased from early to late stages, whereas fungal community diversity decreased, consistent with previous studies [15]. Furthermore, the differences in bacterial community diversity between successional stages were more significant than those in fungal community diversity. The different growth strategies of microbes may be responsible for the significantly decreased bacterial alpha-diversity index. For example, in primary succession, microbial populations live in harsh and unpredictable environments, presenting high reproduction rates and low survival, leading to a high diversity and low abundance. In contrast, in the later stages of the succession, microbial populations live in favorable and predictable environments, presenting low reproduction rates and high survival, leading to increased competitiveness and stable population size [16]. An extensive regional survey revealed that environmental factors are drivers of community diversity and composition [17]. The differences in po plar alpha-diversity in bacteria and fungi communities may be due to microorganisms with different nutritional preferences [18, 19] or bacteria and fungi occupation of different niches [20].

3.2 Changes in Microbial Community Structure with Succession 

Differential abundance analysis was used to study the differences in the microbial community and identify enriched and depleted OTUs. A more significant proportion of enriched and depleted OTUs were found at the initial stages of the bacterial community. In the fungal community, more significant alterations were found at later stages. In addition, we observed that the bacterial community underwent relatively more significant changes than the fungal community, consistent with previous studies [15, 21]. Bacteria and fungi have been shown in several studies to prefer r-strategy and k-strategy groups, respectively [22, 23]. Zhou and co-workers [24] observed that the r-strategy group predominated in early successional forests and that the k-strategy group was widespread in later successional forests. This may result from the faster growth and turnover rates of bacteria compared with fungi during early successional stages [25, 26]. This effect also appears to exist in the succession process of poplar wood. In addition, bacteria can act as primary and secondary wood colonizers, fixing nitrogen, degrading or modifying wood's chemical composition, and providing other microorganisms' nutrient requirements [27]. Moreover, the bacterial community was more variable than the fungal community, probably due to the relatively short duration of the experiment. We further analyzed the changing pattern of microbial taxonomic composition in five successional stages. Bacterial groups, such as Proteobacteria, Actinobacteriota, and Bacteroidota, were dominant bacterial taxa in the five successional stages, consistent with the findings of a growing number of studies in different ecosystems [28, 29]. Notably, the relative abundance of two genera, Sphingomonas and Methylobacteriumbelonging to Proteobacteria increased significantly with succession (Fig. S2A). Previous studies have shown that Sphingomonas has a strong aromatic decomposition ability, degrading aromatic compounds containing methyl groups to methanol and aromatics through demethylation reaction, and can play an essential role in the decomposition of lignin [30, 31]. At the same time, the released methanol act as a carbon source for Methylobacteriumwhich is beneficial to the growth of this bacteria [32]. These findings suggested that Sphingomonas and Methylobacterium play a functional role during the discoloration of poplar wood. 

Ascomycota and Basidiomycota were the main phyla in the poplar for fungal communities among different successional stages, presenting significant abundance changes. This result was consistent with the findings of previous studies [33, 34]. This alteration was mainly due to the significant increase of the genus Sphaeropsis (Ascomycota), considered a crucial contributor to wood discoloration [35, 36]. 

LEfSe analysis was also used to understand microbial communities' variation in different succession stages. Sphaeropsis were enriched at the S4 stage, suggesting that this fungus plays a vital role in the discoloration of poplar wood.

3.3 The different assembly processes underwent by bacterial and fungal communities across successional stages

The fundamental topic of microbial ecology is to uncover the underlying microbial assembly processes [37]. Niche theory states that deterministic factors (i.e., species traits, environmental conditions, and interactions among species) determine the structure of communities [38]. βNTI values were calculated, and RCbray analysis was performed to investigate the role of neutral and niche processes. Results showed that the stochastic processes control the bacterial assembly processes in the early stages, but the stochastic processes decrease with the succession stages, evolving from stochastic to deterministic processes. Similarly, the fungal community was controlled by stochastic processes in the early stages, but in contrast, the assembly process evolves further into stochastic processes during the later successional stages. These findings are consistent with previous studies that assumed that the initial establishment of a community is primarily determined by stochasticity [39]. For example, such studies demonstrated that microbial communities in the early stages of the succession are less similar, being dominated by taxa that can use many different resources, suggesting a strong potential influence of stochasticity [40]. More generally, if a variety of species can evolve effectively in a given environment, stochasticity is likely to dominate the early stages of community formation [41]. Furthermore, the different assemblage processes could be related to the various growth strategies of bacteria and fungi within the succession [24]. Bacterial diversity decreases with succession, while fungal diversity increases with succession. According to a recent study, highly diverse communities are dominated by stochastic processes, while low diverse communities are dominated by deterministic processes that limit community functioning, which is consistent with our findings [42]. RCbray analysis results indicate that variable selection was the dominant ecological process shaping the microbial community among different successional stages, followed by dispersal limitation and undominated process. Variable selection increases with the successional stage. Previous research has shown that communities with low biomass and population size are more susceptible to drift (stochastic processes) or founder effects. However, in an established community with a saturated population or small community size, the local role of stochastic or deterministic processes may be significantly affected by dispersal influenced by the local environment [43]. Dispersal limitation has been widely accepted as an essential driver in shaping microbial community assemblages at different successional stages [44]. The Shannon and Chao1 indices of the bacterial community showed a decreasing trend from S0 to S4. In contrast, the Shannon index of the fungal community showed an increasing trend from S0 to S4, while Chao1 of fungal showed no significant differences between successional stages. These results are consistent with previous research showing that low species diversity in communities could enhance environmental selection and contribute to deterministic (selection) processes followed by dispersal limitation [43]. Another study reported that colonization of specific taxa in a specific environment enhances environmental selection [45]. Due to the regional variance in the selection environment, taxa selected in one area may not be selected in another [46]. For example, new spatially structured ecological niches emerge (e.g., anoxic pockets), and communities are driven toward a 3D architecture [47]. It is expected that increasing environmental variability in this situation will lead to differences in the composition of local communities. As a result, the relative importance of variable selection is expected to increase as succession progresses. In this study, we also noted that some functional taxa (e.g., SphingomonasMethylobacterium, and Sphaeropsis) increased significantly with the successional stage. Furthermore, the undominated process, which accounts for a relatively large proportion of the fungal assembly process, is also present throughout the succession; being difficult to predict the variation of fungal communities within the succession process[48].

Lines 316-374: please consider replacing it with

4. Materials and Methods

4.1 Sample collection and site description

Samples of Nanlin 95 poplar were taken from the city of Suqian (Jiangsu Province, China) and stored at the Jurong wood factory (Zhenjiang, Jiangsu Province, China). Further samples were taken from the stored wood at intervals of 0, 1, 2, 3, and 4 months (numbered S0, S1, S2, S3, and S4respectively). The five-point sampling method collected circular sections of 2.5 cm diameter at positions with uniform discoloration. The entire sample circular section was wiped with a sterile degreasing cotton ball and proceeded for DNA extraction. Three replicates were taken for each sampling point. 

4.2 DNA extraction and high-throughput sequencing

Genomic DNA was extracted using a Fast DNA™ Spin Kit (MP Biomedicals, USA), following the manufacturer's instructions. A NanoDrop 2000 Spectrophotometer (Thermo Scientific, Wilmington, DE, USA) was used to determine the quality of the extracted DNA. 

The bacterial-specific V3-V4 region was amplified with the 338F (5′-ACTCCTACGGGAGGCAGCA-3′) and 806R (5′-GGACTACHVGGGTWTCTAAT-3′). The fungal-specific ITS region was amplified with the ITS1 (GTGAATCATCGARTC) and ITS2 (TCCTCCGCTTATTGAT) primer sets. 

The parameters for the PCR amplification were: 3 min of initial denaturation at 94℃, followed by 24 cycles consisting of 5 seconds of denaturation at 95ºC, 90s of annealing at 57℃, 10s of elongation at 72℃, and a final extension at 72℃ for 5 min.

Pooled PCR products were purified using the GeneJETTM Gel Extraction Kit (Thermo Scientific, USA). Finally, GENEWIZ, Inc (South Plainfield, NJ) sequenced the purified products using an Illumina Miseq platform (Illumina, USA).

4.3 Processing of sequence analysis

The QIIME pipeline (Version 1.9.1) was used to process the microbiome sequences 341 [49]. Reads shorter than 200 bp and with an average base quality score of < 20 were considered low quality and omitted from further analysis. The UCHIME method was used to classify and delete chimeric sequences [50]. Sequences with 97% identity were grouped into operational taxonomic units (OTUs) [51]. An annotated OTU analysis was performed using the Mothur algorithm to find representatives of taxa and count the number of OTUs per sample in the taxonomic information [52]. Bacterial annotation followed the Silva database (https://www.arb-silva.de/), while fungal annotations followed the UNITE database (https://unite.ut.ee/). 

4.4 Statistical and bioinformatics analysis

Bacterial and fungal community alpha-diversity indices (Shannon and Chao1) were calculated using the R package "vegan" [53]. Beta-diversity indices (Bray-Crust dissimilarity) were calculated using QIIME. The R "vegan" package was also used to reconstruct the bacterial and fungal community composition, allowing Principal Coordinates Analysis (PCoA) based on dissimilarity matrices.  

The differential abundance of OTUs was illustrated using Volcano plots, with p-values adjusted according to the false discovery rate (FDR) [54]. The log2 Fold Change (log2FC) and adjusted p-values were calculated using the "DESeq2" R package. The "ggplot2" package was used to construct differential abundance plots for the OTUs. Statistical biomarkers between treatments were identified through Linear discriminant analysis effect size (LEfSe) measurements [55] on the Hutlab Galaxy website application (http://huttenhowe.sph.harvard.edu/galaxy/). 

This study preferentially performed Null-modeling-based approaches to infer community assembly mechanisms [56]. With this approach, the community assembly mechanism can be detected by estimating the observed ecological patterns' standard deviation compared to the randomly shuffled ecological patterns generated by the null model. Generally, variations were investigated by comparing beta-diversity metrics (βNTI values and RCbray). Here, the null model analysis was performed using a framework described by Stegen and co-workers [57] to classify community pairs into underlying drivers of deterministic processes (e.g., homogeneous selection and variable selection) or stochastic processes (e.g., dispersal limitation, homogeneous dispersal, and "non-dominant"). Briefly, The relative influence of dispersal limitation, homogeneous dispersal, and undominated process were quantified as a pairwise comparison between |βNTI| <2, RCbray >0.95, |βNTI| <2, RCbray < -0.95 and |βNTI | <2, -0.95

Lines 375-395: please consider replacing it with

5. Conclusions

This study investigated the variation of poplar wood microbial communities in response to different successional stages. Our results showed that the diversity and composition of microbial communities varied greatly with succession. Bacterial diversity increased, and fungal diversity decreased as the succession proceeded, indicating that bacterial and fungal communities had differential responses to succession. The relative abundance of the dominant phyla was significantly different across the five successional stages: Proteobacteria and Acidobacteria were the most common phyla of bacteria, while Ascomycota and Basidiomycota were the most common phyla of fungus. Notably, the relative abundance of SphingomonasMethylobacterium, and Sphaeropsis significantly increases with the successional stage. The structure of microbial communities showed a separation across the successional stages. Based on the volcano plot analysis, we found that the bacteria experienced more alteration in the early stages of the succession, while the fungi occurred in the late stages. The main ecological processes for microbial community assembly are variable selection, dispersal limitation, and undominated process. Stochastic processes dominate both bacterial and fungal assembly processes in the early stages. In the course of succession, bacterial community assembly processes tend to develop towards deterministic processes, whereas, on the contrary, fungi tend to become more stochastic. In summary, it provides insights into the changes in microbial communities during poplar wood discoloration.

Author Response

Point 1: In the reviewer's opinion, the authors of this paper do not have English as their native tongue. A high degree of editing and revising will be necessary before the manuscript can be appropriately reviewed, as there are numerous statements whose meaning is unclear, repeated sections of the text (sometimes the same text is repeated three times), the is convoluted and hard-to-read. Also, some formatting issues must be carefully corrected (for instance, figure legends). 

Response 1: First of all, we would like to say thank you so much for giving us a chance to revise our manuscript. We gratefully thank the reviewer for taking the time to offer such constructive remarks and suggestions, which have allowed us to improve the quality of the manuscript significantly.We have carefully corrected the language and grammatical errors in this revision. The manuscript was also checked for clarity and accuracy of English by Academic Research Editors Ltd., a company registered in England. In the attachment, we also provide proof of editing.

Point 2: Therefore, due to the extension of the corrections needed, instead of presenting the suggestion by line, the reviewer presents the bulk sections of the text (suggestion in bold). In the Result section, several sentences were removed since they were Discussion and not Results.

Response 2: We would like to thank the reviewer for this constructive feedback. We appreciate the thoughtful and positive comments, which have certainly helped to improve the presentation and quality of our paper. We have revised our paper according to the specific suggestions offered by the reviewer.

Point 3: Lines 9-15: confusing text, please consider rewrite

Response 3: Thank you so much for your valuable comments and suggestions. we are sorry to not made a clear statement in mentioned paragraph. In revised manuscript, we rewrite it carefully.

Point 4: Line 16: please consider replacing it with

 Response 4: Thank you for your rigorous consideration. We have substantially revised this paragraph carefully according to your suggestions.

Point 5: Lines 30-91: please consider replacing it with

 Response 5: We have substantially revised this paragraph carefully according to your suggestions.

Point 6: Lines 92-199: please consider replacing it with

 Response 6: We have substantially revised this paragraph carefully according to your suggestions.

Point 7: Lines 200-315: please consider replacing it with

 Response 7: We have substantially revised this paragraph carefully according to your suggestions.

Point 8: Lines 316-374: please consider replacing it with

 Response 8: We have substantially revised this paragraph carefully according to your suggestions.

Point 9: Lines 375-395: please consider replacing it with

 Response 9: We have substantially revised this paragraph carefully according to your suggestions.
